# Factors related to the development of health-promoting community activities in Spanish primary healthcare: two case–control studies

Sebastià March,[1,2] Joana Ripoll,[1,2] Matilde Jordan Martin,[3]
Edurne Zabaleta-del-Olmo, Carmen Belén Benedé Azagra,[4] Lázaro Elizalde Soto,[5]
Mª Clara Vidal,[1,2] María de Lluc Bauzà Amengual,[6] Trinidad Planas Juan,[7]
Damiana Maria Pérez Mariano,[7] Micaela Llull Sarralde,[8] Juan Luís Ruiz-Giménez,[9]
Rosa Bajo Viñas,[10] Carmen Solano Villarubia,[11] Maria Rodriguez Bajo,[12]
Manuela Cordoba Victoria,[13] Marta Badia Capdevila,[14] Elena Serrano Ferrandez,[14]
Maria Bosom Diumenjo,[15] Isabel Montaner-Gomis,[16] Buenaventura Bolibar-Ribas,[14]
Angel Antoñanzas Lombarte,[17] Samantha Bregel Cotaina,[18] Ana Calvo Tocado,[19]
Barbara Olivan Blázquez,[18] Rosa Magallon Botaya,[20] Pilar Marín Palacios,[5]
Margarita Echauri Ozcoidi,[5] María Jose Perez - arauta,[5] Joan Llobera,[1,2]
Maria Ramos[2,21]

JR and MLBA contributed equally.

For numbered affiliations see end of article.

**Correspondence to**
Sebastià March;
smarch@ibsalut.caib.es

## ABSTRACT

**Objective** Spanish primary healthcare teams have the responsibility of performing health-promoting community activities (CAs), although such activities are not widespread. Our aim was to identify the factors related to participation in those activities.

**Design** Two case–control studies.

**Setting** Performed in primary care of five Spanish regions.

**Subjects** In the first study, cases were teams that performed health-promoting CAs and controls were those that did not. In the second study (on case teams from the first study), cases were professionals who developed these activities and controls were those who did not.

**Main outcome measures** Team, professional and community characteristics collected through questionnaires (team managers/professionals) and from secondary sources.

**Results** The first study examined 203 teams (103 cases, 100 controls). Adjusted factors associated with performing CAs were percentage of nurses (OR 1.07, 95% CI 1.01 to 1.14), community socioeconomic status (higher vs lower OR 2.16, 95% CI 1.18 to 3.95) and performing undergraduate training (OR 0.44, 95% CI 0.21 to 0.93). In the second study, 597 professionals responded (254 cases, 343 controls). Adjusted factors were professional classification (physicians do fewer activities than nurses and social workers do more), training in CAs (OR 1.9, 95% CI 1.2 to 3.1), team support (OR 2.9, 95% CI 1.5 to 5.7), seniority (OR 1.06, 95% CI 1.03 to 1.09), nursing tutor (OR 2.0, 95% CI 1.1 to 3.5), motivation (OR 3.7, 95% CI 1.8 to 7.5), collaboration with non-governmental organisations (OR 1.9, 95% CI 1.2 to 3.1) and participation in neighbourhood activities (OR 3.1, 95% CI 1.9 to 5.1).

**Conclusions** Professional personal characteristics, such as social sensitivity, profession, to feel team support or

### Strengths and limitations of this study

► This study examined a topic relatively unexplored, community health promotion by primary healthcare teams.

► A case–control approach was chosen, where cases were subjects (professionals or teams) who were developing health-promoting community activities (CAs), and controls those who did not.

► Design allows exploration of many distinct factors simultaneously, from different sources, and in two levels: health team and professional.

► An operative definition of CAs was used, which recognises the heterogeneity of health promotion interventions developed with the community in Spanish primary healthcare settings

► Because of its observational design, there are limitations to establishing causal inferences.

motivation, have influence in performing health-promoting CAs. In contrast to the opinion expressed by many professionals, workload is not related to performance of health-promoting CAs.

## INTRODUCTION

The Ottawa Charter for Health Promotion[1] defined health promotion as the process of enabling people to increase control over, and to improve, their health. This definition implies the importance of empowering people[2] to take control over what determines their health. One tool to work in this direction

are health-promoting community activities (CAs). These can come in a variety of forms, from walking groups in a neighbourhood to holding meetings with local authorities, schools or civil associations to develop specific health programmes. Experts at numerous international conferences have agreed on the importance of health-promoting CAs for improving public health,[1 3 4] and there is some evidence that these activities are effective.[5]

Since the implementation of healthcare reform in Spain during the 1980s,[6 7] primary healthcare teams (PHTs) have had responsibility for carrying out health promotion activities. These teams, located in healthcare centres throughout the country, are composed of physicians, community nurses, paediatricians, midwives, social workers and other healthcare professionals (gynaecologists, psychologists and psychiatrists) in a system that offers almost-universal and free healthcare. These conditions are ideal for an interdisciplinary approach to encourage health-promoting CAs. However, as in other countries,[8] implementation of these interventions differs greatly among the different healthcare teams,[9–11] and relatively few professionals are involved.[11–13]

There has been limited studies of the factors related to the development of health-promoting interventions in the community. These factors may be broadly classified as those related to the community, to the team or to the individual professionals. With regard to community factors, some studies indicated that the rural/urban environment,[14] size of the municipality,[15] level of social interaction within the community[15–17] and the presence of a focal point for participation in community health initiatives[15 16] influence the practice of health-promoting CAs. The PHT factors include job satisfaction,[18] management of time and setting within the centre,[14] composition and organisation of the team,[15 16] years of team operation,[15] intrateam support[17] and work burden.[8 19] Studies of professionals have highlighted sex,[19–21] age,[15 20] professional status,[14] specific training,[22–25] motivation[17 18] and the model applied (biomedical or psychosocial).[18] The attitudes and beliefs of the professionals, such as trust in the effectiveness of preventive actions,[18 22 25–27] self-reported efficacy in carrying out activities[8 22 23] or to support community participation,[15 21] are also important. An additional element is professionals' social sensitivity, as evaluated through their personal political leanings.[28] According to professionals themselves, the most important factor is the lack of time available to address the care burden.[19 22 24–27] All these features have been observed separately, mostly in descriptive studies.

This study presents some of the results of the Factors Related to Health-Promoting Community Activity Development in Primary Care (frAC) Project, whose complete methodology and aims were described elsewhere.[29] The purpose of this study is to identify PHT and community factors related to implementation of health-promoting CAs in primary care and factors that may explain why only some professionals within a team participate in those CAs.

## METHODS

Two case–control studies were performed. In the first, the cases were the teams that performed health-promoting CAs and the controls were those that did not. The second study examined only teams that implemented these CAs to adjust for community/team factors; in this second analysis, the cases were professionals who developed these activities and the controls were those who did not. The study was set in primary healthcare in five regions of Spain (Balearic Islands, Catalonia, Aragon, Madrid and Navarra). Data collection was carried out between 2009 and 2010.

### Selection of cases and controls

We used a conceptual definition of health-promoting CA reached in a previous study.[30 31] This was converted into an algorithm for use in this study[29]: this activity had to be a non-isolated activity carried out in the previous year, in which the professional participated on behalf of the health centre. Furthermore, it had to have involvement and active participation of the community, or the population had to have the capacity to influence the intervention, or it was a cross-sector activity involving collaboration with entities outside of healthcare (eg, education, social services). Health preventive interventions which does not have a health promotion approach were excluded.

We telephoned all the health centres in the research areas asking if they had participated in health-promoting CAs during the last year. If they did, we contacted a participant of the activity and gave him/her a questionnaire to confirm that the activity met our inclusion criteria. For the first study, health teams that participated in at least one CA in the last year were selected as cases, and controls were those who had not participated. For the second study, we selected all professionals who participated in the implementation of those CAs as cases and professionals who did not participate as controls. In every health team, we selected at least one control for every case.

### Variables

Information was gathered through three distinct questionnaires (available on request) that were given to the community, the team and the professionals.[29] Description of health-promoting CAs is described elsewhere.[30]

The data collected from the community (from questionnaires to centre managers and secondary sources) were: demographic composition; socioeconomic level; degree of social interaction; health centre setting; geographical dispersion based on an ordinal variable used by the National Health System for resource allocation (from G1 (less dispersed) to G4 (more dispersed)) and existence of health boards (community participation agencies described in health legislation).[16]

The data collected from the team (collected from secondary sources and questionnaires to centre managers) were: percentages of different professionals (nurses, physicians and paediatricians); year of opening; population of service area; professional/population

and doctor/nurse ratios; availability of a space in the centre to carry out group activities and the presence of a team member responsible for health education; collaboration with the area health board where applicable; training of resident physicians; nurses and nursing or physiotherapy students and evaluation of professional relationships between distinct professional disciplines (based on a Likert-type scale ranging from 1 (poor) to 5 (excellent)).

The data collected from the professionals (from individual self-administered questionnaires) were: sex; age; profession; working situation; healthcare burden; size of quota allocated; average number and duration of consultations per day at the centre or at home; tutoring; participation in research or specific training in health promotion in the previous 5 years; years since graduation; years worked in primary care and as a member of the current healthcare team; autonomy to organise scheduling; arrangements with colleagues to cover healthcare tasks while carrying out CAs; support of fellow team members in conducting these activities; working atmosphere; job satisfaction and self-assessed efficacy in conducting community work (based on a Likert-type scale ranging from 1 (poor) to 5 (excellent)).

Opinions and attitudes were measured by responses to various statements developed by the research team, using a Likert-type scale with four options ranging from 'strongly agree' to 'strongly disagree'. The statements dealt with trust in the effectiveness of the health promotion CAs, motivation to carry out these activities, citizen participation in healthcare decision-making, need to strengthen agencies involved in citizen participation in healthcare, responsibility for primary care in the community, definition of professional role, link between health education and community participation and the effectiveness of the boards in driving CAs. A four-item validated scale was used to evaluate each professional's clinical practice style (biomedical–psychosocial)[32 33] by assessing the attitude towards the demands of psychosocial and clinical care at work. Finally, social sensitivity data were collected through proxy variables: participation during the previous 12 months in activities in the neighbourhood where the professional works; involvement in any non-governmental organisation or civil association and political tendency (assessed through a visual scale adapted from the Centre for Sociological Research[34] Spanish monthly barometer).

### Calculation of sample size

For the first study, we calculated that at least 91 teams would be needed in each group based on an OR of 2.5, an alpha risk of 0.05, a beta of 0.2 and an expected proportion of 0.5 in the controls. For the second study, we calculated that at least 222 professionals would be needed in each group based on an OR of 1.75, an alpha risk of 0.05, a beta of 0.2 and an expected proportion of 0.5 in the controls.

### Data analysis and management

A descriptive analysis of all variables was carried out to compare their distributions in each group. Categorical variables are expressed as percentages and continuous variables as means with 95% CIs or as medians and percentiles depending on the distribution. Main variable was, for both studies, to be case or control. The relationship with the main variable was assessed using the $\chi^2$ test for categorical variables and Student's t-test and the Mann-Whitney U non-parametric test for continuous variables. The strength of an association was expressed through unadjusted OR and its 95% CI.

We also performed a logistic regression analysis to calculate adjusted ORs. We constructed a first model using each variable whose p value was less than 0.2 in the bivariate analysis. From this model, we tested and compared variations using the maximum likelihood method, eliminating variables that had p values less than 0.05 in the Wald test. The collinearity of the variables was examined and interactions were tested. We established a different model for the healthcare teams and for the professionals. All analyses were conducted using SPSS V.14 and Epidat V.3.1 software.

### Ethical aspects

The questionnaire was completed anonymously by the professionals, with the identity of individuals and centres coded to ensure blinding of the researchers.

## RESULTS
### Factors related to the healthcare team

We examined 203 healthcare teams (103 cases (50.7%) and 100 controls) in the first study. Table 1 shows the results of the initial bivariate analysis. The variables significantly related to team involvement in CAs were the socioeconomic level of the community, collaboration on training in physiotherapy or nursing, participation in a health board, percentage of nurses in the team, patient/physician ratio and patient/nurse ratio.

Eleven teams (5.0%) were excluded from the logistic regression analysis due to missing data. The three variables remaining in the final model were percentage of nurses in the team (OR 1.07 for every 1% increase; 95% CI 1.01 to 1.14); socioeconomic level of the community (high and medium-high vs low and medium-low; OR 2.16; 95% CI 1.18 to 3.95) and having undergraduate training at the healthcare centre (OR 0.44; 95% CI 0.21 to 0.93). Nagelkerke's $R^2$ was 0.11.

### Factors related to professionals

A total of 597 professionals responded to the questionnaire, 254 (42.4%) cases and 343 (57.5%) controls. Ninety-six professionals (62 cases and 34 controls) refused to participate. Participants were 298 nurses (52.4%), 191 doctors (33.6%), 58 paediatricians (10.2%) and 22 social workers (3.8%). Tables 2 and 3 show the results of the bivariate descriptive analysis. Table 2 shows that the

**Table 1** Relationship of community and primary healthcare team (PHT) characteristics with performance of health-promoting activities in the community

| Nominal variables | PHT cases n (%) | PHT controls n (%) | Unadjusted OR (95% CI) |
|---|---|---|---|
| Healthcare setting | | | |
| Urban | 73 (71.6) | 68 (68.0) | Ref. |
| Rural | 29 (28.4) | 32 (32.0) | 0.84 (0.46 to 1.54) |
| Socioeconomic level | | | |
| High and medium-high | 52 (51) | 65 (65) | 1 |
| Medium-low and low | 50 (49) | 35 (35) | **1.79 (1.01 to 3.14)** |
| Degree of social interaction | | | |
| High | 22 (23.9) | 23 (26.1) | Ref. |
| Medium | 43 (46.7) | 35 (39.8) | 1.28 (0.61 to 2.68) |
| Low | 27 (29.3) | 30 (34.1) | 0.94 (0.43 to 2.06) |
| Population of municipality | | | |
| >1 million | 47 (45.6) | 48 (48) | Ref. |
| 500 000–1 million | 7 (12.6) | 7 (7) | 1.90 (0.70 to 5.18) |
| 10 000–500 000 | 22 (21.4) | 33 (33) | 0.68 (0.35 to 1.33) |
| <10 000 | 21 (20.4) | 12 (12) | 1.79 (0.79 to 4.04) |
| Degree of geographical dispersion | | | |
| G1 (less dispersed) | 62 (66.7) | 60 (63.8) | Ref. |
| G2 | 11 (11.8) | 17 (18.1) | 0.63 (0.27 to 1.45) |
| G3 | 9 (9.7) | 11 (11.7) | 0.79 (0.31 to 2.05) |
| G4 (more dispersed) | 11 (11.8) | 6 (6.4) | 1.77 (0.67 to 5.10) |
| Active health board | 67 (66.3) | 60 (61.9) | 1.21 (0.68 to 2.17) |
| Training centre | | | |
| Resident doctors | 35 (34.7) | 34 (34.3) | 1.01 (0.56 to 1.82) |
| Resident nurses | 23 (23.5) | 29 (30.2) | 0.71 (0.37 to 1.34) |
| Nursing/physiotherapy students | 73 (72.3) | 84 (85.7) | **0.43 (0.21 to 0.89)** |
| Space for group activities | 83 (81.4) | 79 (79) | 1.16 (0.58 to 2.32) |
| Health education manager | 45 (44.1) | 46 (46) | 0.93 (0.53 to 1.61) |
| Participation in health board | 56 (83.6) | 38 (63.3) | **2.95 (1.28 to 6.78)** |
| **Continuous variables** | **Mean (95% CI)** | | **p**[*] |
| PHT operational years | 14.3 (13 to 15.2) | 12.9 (11.7 to 14.1) | 0.075 |
| Total no. professionals | 36.4 (33.1 to 39.6) | 37.5 (34 to 41.1) | 0.486 |
| % nurses | 35.4 (34.5 to 36.4) | 33.5 (32.4 to 34.5) | **0.010** |
| % paediatricians | 7.0 (6.3 to 7.7) | 7.4 (6.7 to 8.1) | 0.450 |
| Population covered by PHT | 20 031 (18 059 to 22 002) | 21 770.4 (19 530 to 24 010) | 0.247 |
| | **Median (IQR)** | | **p**[**] |
| % doctors | 33.3 (29.5–38.3) | 33.3 (29.9–36.4) | 0.532 |
| % non-European Union immigrants | 11.1 (5.8–16.2) | 11.1 (8.6–15.7) | 0.690 |
| % population<15 years | 13.0 (11.6–14.7) | 12.9 (11.3–14.4) | 0.352 |
| % population>65 years | 17.8 (15.0–21.8) | 18.9 (13.8–21.5) | 0.976 |
| % population women | 51 (49.5–53) | 51.2 (50.3–53.1) | 0.403 |
| Population ratio registered/professional | 524 (444–658) | 578 (482–669) | 0.122 |
| Population ratio registered/doctor | 1659 (1454–1905) | 1734 (1533–2040) | **0.034** |
| Population ratio registered/nurse | 1508 (1284–1847) | 1723 (1434–2057) | **0.008** |

**Table 1** Continued

| Nominal variables | PHT cases n (%) | PHT controls n (%) | Unadjusted OR (95% CI) |
|---|---|---|---|
| Doctor/nurse ratio | 0.94 (0.86–1.09) | 1 (0.89–1.14) | 0.085 |
| **Professional relationships** | | | |
| Doctor::nurse | 4 (3–4) | 4 (3–4) | 0.376 |
| Doctor::social worker | 4 (3–5) | 4 (3–4) | 0.978 |
| Nurse::social worker | 4 (3–5) | 4 (3–4) | 0.845 |

Data from Spain 2009. For dichotomous variables, the reference value is always absence of the variable. *Student's t-test p. **Mann-Whitney U test p. Boldface: p≤0.05.

variables related to involvement in health-promoting CAs were sex, profession, training, self-confidence and work situation. Also, years of experience, age, workload, practice style and political orientation seem to be related. Table 3 summarises the responses of the professionals' opinions and attitudes to health promotion, community participation and social sensitivity.

Due to missing data, 100 (16.8%) professionals were excluded from the multivariate analysis, leaving a total of 497. There were no significant differences between the excluded and included individuals. The final multivariate model (table 4) shows that the following factors were significantly associated with health-promoting CAs: profession (social workers do the most, followed by nurses and then physicians); specific training in CAs; motivation; social sensitivity (collaboration with NGOs and participation in neighbourhood activities); perception of support and coverage by colleagues while doing CAs; years working in the healthcare team and being a nursing student tutor.

## DISCUSSION

Our study of the participation of healthcare workers in health-promoting activities in the community suggests that factors of the individual healthcare professional were more important than those of the healthcare team or the community. In particular, the specific profession, attitudes and opinions of an individual seemed to be crucial. Professional workload, which was one of the more mentioned factors by the professionals in others studies, does not have any impact when adjusted by other individual factors. These results confirm the frAC Project[29] hypothesis and those of other authors[21 28]

Our assessment of variables related with healthcare team participation in health-promoting CAs suggests that team characteristics had greater impact than community characteristics. One exception was the socioeconomic level of the community. This makes sense in the context of the Spanish healthcare system, which attempts to offer equitable care to people according to their needs and community interventions are important tools to reduce inequalities in healthcare.[35] It should be noted that we found no relationship between the level of social

interaction in the community and the implementation of CAs, in contrast with the findings of other authors.[16 17]

Our results showing the important role of nurses in health-promoting CAs are consistent with those of other authors[13 14 17 36] who highlighted the role of nurses in the development of CAs. This result is expected because nurses usually receive more training in community care. Another crucial group is social workers. Previous research reported the importance of social workers,[13] but their role in the community varies in different regions of Spain. Their presence seems fundamental for development of health-promoting CAs in regions where they have significant roles in the community. In Spain, community work is generally not provided as part of primary healthcare; instead, it is relegated to professional volunteers or specific programme. Thus, our results recognise the importance of work by nurses and social workers and point to the possibility that their roles could be modified or clarified. We also found that health centres that train undergraduate nurses have a lower level of CAs, but that professional nurse trainers have a greater involvement in health-promoting CAs. This apparent contradiction may be resolved by considering two factors. First, the variable at the health centre level included training in various disciplines (from nursing to physiotherapy) without distinction, even though there is clearly difference among these in communitarian tradition. Second, the need for increased training would make it difficult for the health team to engage in CAs, although among the cases, trainers carried out more activities than others. Healthcare workers who worked for more years at a centre were also more likely to perform CAs. This may be because medium-term and long-term projects require continuity and stability, and it is more difficult for professionals who continually change centres or have temporary contracts to participate in health-promoting CAs.

The pressures of working in healthcare and the heavy work burden were the main arguments offered by professionals when asked about their low participation in health promotion and prevention activities.[19 22 24–27] Our bivariate analysis supported this relationship, although the multivariate analysis did not. This implies that the professionals' perception was incorrect. In fact, the pressure

**Table 2** Relationship of professional variables with performance of health-promoting community activities (CAs)

| Nominal variables | Cases n (%) | Controls n (%) | Unadjusted OR (95% CI)* |
|---|---|---|---|
| Sex | | | |
| Male | 41 (16.4) | 84 (25.3) | Ref. |
| Female | 209 (83.6) | 248 (74.7) | **1.72 (1.13 to 2.61)** |
| Profession | | | |
| Nurse | 156 (64.7) | 142 (43.3) | Ref. |
| Doctor | 40 (16.6) | 151 (46) | **0.24 (0.16 to 0.36)** |
| Paediatrician | 24 (10) | 34 (10.4) | 0.64 (0.36 to 1.3) |
| Social worker | 21 (8.7) | 1 (0.3) | **19.11 (2.53 to 143.9)** |
| Specialised general practitioner | 22 (55) | 91 (60.3) | 0.81 (0.39 to 1.62) |
| Tutor of resident doctors | 10 (25) | 37 (24.5) | 1.02 (0.45 to 2.3) |
| Nursing tutors | 93 (59.6) | 58 (40.8) | **2.13 (1.34 to 3.39)** |
| Work situation | | | |
| Permanent | 204 (80.3) | 238 (70) | Ref. |
| Temporary contracts | 40 (15.7) | 74 (21.8) | **0.63 (0.41 to 0.96)** |
| Other | 10 (3.9) | 28 (8.2) | **0.41 (0.19 to 0.87)** |
| Participation in research projects in the last 5 years | 115 (47.7) | 147 (43.9) | 1.16 (0.83 to 1.62) |
| Specific training in CAs | 142 (58.2) | 104 (31.7) | **2.99 (2.12 to 4.23)** |
| Job satisfaction | | | |
| None/little | 18 (7.1) | 34 (10) | Ref. |
| Average | 64 (25.3) | 129 (38.1) | 0.93 (0.49 to 1.78) |
| Quite a lot/a lot | 171 (67.5) | 176 (51.9) | 1.83 (0.99 to 3.37) |
| Working environment satisfaction | | | |
| None/little | 14 (5.6) | 33 (9.8) | Ref. |
| Average | 82 (32.5) | 103 (30.5) | 1.87 (0.94 to 3.73) |
| Quite a lot/a lot | 156 (61.9) | 202 (59.8) | 1.82 (0.94 to 3.51) |
| Self-confidence to carry out CAs | | | |
| None/little | 31 (12.3) | 72 (21.6) | Ref. |
| Average | 81 (32.1) | 154 (46.2) | 1.22 (0.74 to 2.01) |
| Quite a lot/a lot | 140 (55.5) | 107 (32.1) | **3.03 (1.86 to 4.96)** |
| **Continuous variables** | Median (p25–p75) | | p* |
| Age | 49 (42–54) | 48 (39–53) | **0.048** |
| Years qualified | 25 (18–31) | 24 (15–29) | **0.024** |
| Years in primary care | 19 (11–23) | 17 (10–23) | 0.552 |
| Years in current health centre | 11 (4–18) | 7 (2–15) | **0.000** |
| Size of assigned quota | 1737 (1300–2043) | 1600 (1300–1900) | **0.026** |
| Average daily consultations | 20 (15–30) | 30 (20–35) | **0.000** |
| Average daily home visits | 1 (0.5–3) | 1 (1–2) | 0.932 |
| Average consultation time | 10 (10–15) | 10 (7–12) | **0.000** |
| Average home visit time | 25 (20–30) | 20 (15–30) | **0.026** |
| Biomedical practice style (+) psychosocial (–) | 8 (6–9) | 8 (7–9) | **0.040** |
| Political orientation: left (–) right (+) | 2 (1–5) | 3 (2–5) | **0.026** |

Data from Spain 2009. For dichotomous variables, the reference value is always absence of the variable. Boldface: $\chi^2$ p≤0.05. *Mann-Whitney U test.

of working in healthcare and the heavy work burden are not associated with low participation in health-promoting CAs.

Our results indicated the importance of the opinions and attitudes of professionals when performing health-promoting CAs, in accordance with other studies.[15 21 22] In fact, all variables regarding the opinions and attitudes of healthcare workers were significantly different between cases and controls. This suggests that individuals who perform CAs have similar attitudes: they employ a biopsychosocial[12 37] practice model, are more oriented towards health promotion, show recognition of the need for citizen participation in health services and are more left-leaning in ideology. We also found that social sensitivity was an

**Table 3** Professional attitudes in cases and controls regarding health-promoting community activities (CAs).

| Variable | Cases n (%) | Controls n (%) | Unadjusted OR (95% CI) |
|---|---|---|---|
| Confidence in CA efficacy | | | |
| Disagree/strongly disagree | 9 (3.6) | 19 (5.7) | Ref. |
| Agree | 110 (43.8) | 196 (59.0) | 1.18 (0.51 to 2.71) |
| Strongly agree | 132 (52.6) | 117 (35.2) | **2.38 (1.03 to 5.46)** |
| Motivated to carry out CAs | | | |
| Disagree–strongly disagree | 44 (17.5) | 149 (45.4) | Ref. |
| Agree | 113 (45.0) | 131 (39.9) | **2.92 (1.91 to 4.44)** |
| Strongly agree | 94 (37.5) | 48 (14.6) | **6.63 (4.08 to 10.7)** |
| Citizen participation in health | | | |
| Disagree/strongly disagree | 23 (9.2) | 46 (14.0) | Ref. |
| Agree | 122 (49.0) | 191 (58.1) | 1.27 (0.73 to 2.21) |
| Strongly agree | 104 (41.8) | 92 (28.0) | **2.26 (1.27 to 4.01)** |
| Citizen participation in health services | | | |
| Disagree/strongly disagree | 24 (9.6) | 48 (14.5) | Ref. |
| Agree | 109 (43.6) | 198 (59.8) | 1.10 (0.64 to 1.89) |
| Strongly agree | 117 (46.8) | 85 (25.7) | **2.75 (1.56 to 4.83)** |
| Professionals' role in the community is well defined | | | |
| Disagree/strongly disagree | 126 (51) | 223 (66.8) | Ref. |
| Agree | 103 (41.7) | 97 (29) | **1.87 (1.32 to 2.67)** |
| Strongly agree | 18 (7.3) | 14 (4.2) | **2.27 (1.09 to 4.73)** |
| Health education and community participation are closely linked | | | |
| Disagree/strongly disagree | 20 (8.1) | 39 (11.7) | Ref. |
| Agree | 128 (51.6) | 194 (58.4) | 1.28 (0.71 to 2.31) |
| Strongly disagree | 100 (40.3) | 99 (29.8) | **1.97 (1.07 to 3.61)** |
| The health board drives CAs | | | |
| Disagree/strongly disagree | 80 (33.8) | 125 (39.8) | Ref. |
| Agree | 123 (51.9) | 163 (51.9) | 1.17 (0.81 to 1.69) |
| Strongly agree | 34 (14.3) | 26 (8.3) | **2.04 (1.14 to 3.65)** |
| Participates in activities in the neighbourhood where the professional is working | 138 (55.0) | 83 (24.6) | **3.75 (2.64 to 5.33)** |
| Collaborates with an Non-Governmental Organisation or civic entity | 151 (59.9) | 133 (39.7) | **2.27 (1.62 to 3.17)** |
| Freedom to organise timetable | 163 (65.7) | 157 (47.7) | **2.10 (1.49 to 2.95)** |
| Possibility colleagues will cover while performing CAs | 99 (40.6) | 79 (23.9) | **2.16 (1.51 to 3.11)** |
| Can depend on colleagues to collaborate on CAs | 215 (86.3) | 222 (67.7) | **3.02 (1.96 to 4.63)** |

Data from Spain 2009. Boldface: $\chi^2$ p≤0.05.

important variable in the model for professionals. This variable measures involvement of the professional with the community beyond consultation work. It is not clear how some practices, responsibility for which is stipulated in primary care legislation and which should be similar for all professionals to assure fairness within the system, can be affected by a professional's personal discourse. Most healthcare professionals whom we interviewed—both cases and controls—said that their community roles were not well defined. This suggests that greater priority should be given

to develop health-promoting CAs in primary healthcare and to clarify the role and responsibility of every professional about this question.

We found that having support within the healthcare team is associated with performing health-promoting CAs. This finding should be highlighted, because all the professionals in this study were selected from the centres that engage in CAs so that, presumably, all cases can count on some support from their centres. It may be that some professionals were unaware of what their colleagues were doing or that there

**Table 4** Multivariate model of performing health-promoting community activities (CAs)

| Variable | General model (n=497)<br>OR (95% CI) |
|---|---|
| Profession | |
| Nurse | Ref. |
| Doctor | 0.44 (0.24 to 0.82) |
| Paediatrician | 1.55 (0.69 to 3.46) |
| Social worker | 25.26 (2.53 to 251.9) |
| Specific training in CAs | 1.91 (1.17 to 3.11) |
| Collaboration with NGO or civic entity | 1.91 (1.17 to 3.13) |
| Possibility of work covered by colleagues | 1.80 (1.08 to 2.99) |
| Support from colleagues in doing CAs | 2.92 (1.49 to 5.73) |
| Motivated to perform CAs | |
| Disagree or strongly disagree | Ref. |
| Agree | 2.29 (1.28 to 4.08) |
| Strongly disagree | 3.74 (1.85 to 7.54) |
| Participates in activities in the neighbourhood | 3.06 (1.86 to 5.05) |
| Years at health centre* | 1.06 (1.03 to 1.09) |
| Nursing student tutor | 1.96 (1.09 to 3.53) |
| Model validity | Value |
| $\chi^2$ | 221.338 |
| Nagelkerke $R^2$ | 0.483 |

*Continuous variable.

were barriers to working with particular colleagues. Regardless of the explanation, this result suggests that application of the teamwork model,[38] an assumed key feature of primary healthcare, has certain weaknesses.

### Strengths and limitations

This study had an observational design, so we could not definitively establish causal relationships. Similarly, our definition of 'health-promoting CA' may be a limitation because this concept can be defined or perceived in different ways; we made an attempt to control for this by use of a definition generated by expert consensus, which we then transformed into an algorithm. The questionnaires we used were ad hoc and some of the data regarding team variables, such as degree of social interaction in the community and the relationship between professionals, were provided by team managers. This may have led to a bias in information gathering due to a lack of knowledge or other unknown factors. Some variables that we did not examine could have explained participation in CAs, such as support from primary care management, the healthcare administration and the centres' own coordinators or nursing managers. It would be valuable to assess the effect of these variables in future studies of the effect of team activities and priorities.

The main strengths of this study are that it examined a facet of healthcare that is relatively unexplored; the study subjects were from five different regions of Spain; the sample size was large and it considered many distinct factors simultaneously. Furthermore, our design of the second study adjusted the effect of team and community variables matching cases and controls by teams, so a multilevel analysis was not necessary. Given the extension of the achieved sample size and the inclusion of regions with different sizes, our results could be generalised to Spanish primary healthcare.

### Implications for healthcare practice

Healthcare administrators who want to develop health promotion activities for the community must provide visible and formal support for community initiatives, define better the expected communitarian tasks of each health professional and provide job stability for all participating workers. Specific training is also important for the formation of health-promoting CAs. It may seem that attitudes, social sensitivity or personal opinions are not modifiable; however, in our opinion, it is possible to modify these factors during undergraduate education. So, community perspective and their associated responsibility should be included in healthcare workers training.

### CONCLUSIONS

A key responsibility of primary healthcare is the provision of health-promoting CAs, although not every team or professional makes them. We describe some factors related with this implication, such as to work in a low socioeconomic level community or to have specific training, support, motivation, social sensitivity or more years in the health team. There were also differences among professional categories, highlighting nurses' and social workers' contributions. The pressures perceived by the professionals were not associated with involvement in CAs.

**Author affiliations**
[1]Primary Care Research Unit of Mallorca, Baleares Health Services-IB-Salut, Palma, Spain
[2]Balearic Islands Health Research Institute (IdISBA), Palma, Spain
[3]Estrecho de Correa Primary Health Center, Madrid Health Services, Madrid, Spain
[4]Oliver Primary Health Center, Aragon Health Services, Zaragoza, Spain
[5]Navarra Public Health Institute, Public Health and Epidemiology CIBER, Pamplona, Spain
[6]Balearic Islands University, Palma de Mallorca, Spain
[7]Son Gotleu Primary Health Center, Baleares Health Services-IB-Salut, Palma de Mallorca, Spain
[8]San Agustí Primary Health Center, Baleares Health Services-IB-Salut, Palma de Mallorca, Spain
[9]Vicente Soldevilla Primary Health Center, Madrid Health Services, Madrid, Spain
[10]Loeches Primary Health Center, Madrid Health Services, Madrid, Spain
[11]DA Sureste Primary Health Center, Madrid Health Services, Madrid, Spain
[12]Freelance, Madrid, Spain

[13]SOMAMFYC, Madrid, Spain

[14]Institut Universitari d'Investigació en Atenció Primària Jordi Gol (IDIAP Jordi Gol), Barcelona, Spain

[15]Sant Rafael Primary Health Center, Àmbit d'Atenció Primària Barcelona-Ciutat, Institut Català de la Salut, Barcelona, Spain

[16]El Carmel Primary Health Center, Àmbit d'Atenció Primària Barcelona-Ciutat, Institut Català de la Salut, Barcelona, Spain

[17]Delicias del Sur Primary Health Center, Aragon Health Services, Zaragoza, Spain
[18]Primary Care Research Unit, Aragon, Spain
[19]Zaragoza Public Health Department, Institut Catala De La Salut, Zaragoza, Spain
[20]Arrabal Primary Health Center, Aragon Health Services, Zaragoza, Spain
[21]Public Health Department, Balearic Islands Health Department, Zaragoza, Spain

**Acknowledgements** We want to express thanks to Magdalena Esteva for her patient reading of the manuscript.

**Contributors** SM was the lead investigator and was responsible for conception and design of the study, analysing data and writing the manuscript. MR has the original idea of the study. SM, JR, JLR-G, IM-G, CBBA and LES coordinated the field work in everyone of the regions involved. All the authors contributed to the design and development of the study, have read the manuscript critically, made contributions and approved the final version.

**Funding** This study was supported by a grant from the Carlos III Health Institute (PI07/90383; PI07/90925; PI07/90636). It also received support from the Health Promotion and Preventive Activities in Primary Health Care Research Network (IAPP network RD 06/0018/ and R12/0005/0011), the European Regional Development Fund and Balearic Islands Research Institute (IdisBA). SM was contracted with the aid of a grant to stabilise employment in health research from the Carlos III Health Institute and benefited from a grant for further studies from the same institution. All institutions are public and did not have any role in study design, analysis or publication.

**Competing interests** All authors have completed the ICMJE uniform disclosure form at www.icmje.org/coi_disclosure.pdf and declare no financial relationships with any organisations that might have an interest in the submitted work in the previous 3 years; no other relationships or activities that could appear to have influenced the submitted work.

**Patient consent** Study with health professionals with aggregated data that remain anonymous. The editors and reviewers have seen the detailed information available and are satisfied that the information backs up the case the authors are making.

**Ethics approval** The study was approved by the Research Commissions in all the participant regions, the research ethics committee in Madrid (Area 1) and the University Research Institute in Primary Care (IDIAP) Jordi Gol in Barcelona.

**Provenance and peer review** Not commissioned; externally peer reviewed.

**Data sharing statement** Questionnaires and anonymised data are available on demand.

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
