## [Reviewer comments · BMJ Open]

ARTICLE DETAILS

TITLE (PROVISIONAL)	Factors related to the development of health-promoting community activities in Spanish primary health care: Two case-control studies
AUTHORS	March, Sebastià; Ripoll, Joana; Jordan Martin, Matilde Jordan Martin; Zabaleta del Olmo, Nieves; Benedé Azagra, Carmen Belén; Elizalde Soto, Lázaro; Vidal, M ^a Clara; Bauzà Amengual, M. de Lluc; Planas Juan, Trinidad; Maria Pèrez, Damiana Maria Pèrez; Llull Sarralde, Micaela; Ruiz-Giménez, Juan Luís; Bajo Viñas, Rosa; Solano Villarubia, Carmen Solano Villarubia; Rodriguez Bajo, Maria; Cordoba Victoria, Manuela Cordoba Victoria; Badia Capdevila, Marta; Serrano Ferrandez, Elena; Bosom Diumenjo, Maria; Montaner-Gomis, Isabel; Bolibar-Ribas, Buenaventura; Antoñanzas Lombarte, Angel; Bregel Cotaina, Samantha; Calvo Tocado, Ana; Oliván Blázquez, Barbara; Magallon Botaya, Rosa; Marín Palacios, Pilar Marín Palacios; Echaury Ozcoidi, Margarita; Perez-Jarauta, M ^a Jose Perez-Jarauta; Llobera, Joan; Ramos, Maria

VERSION 1 - REVIEW

REVIEWER	Alcalde-Rabanal J Institute of Public Health, Mexico
REVIEW RETURNED	07-Feb-2017

GENERAL COMMENTS	Title Factors related to the development of health-promoting community activities in Spanish primary health care: Two case-control studies I am not sure if it is the correct title, because in the paper I can't find a definition that the author use to classify community activities. If health personnel visit communities to follow a person who live with diabetes, it is not health promotion. This activity is preventive. In my opinion is important to describe with better detail community activities in order to verify if this paper is related with preventive framework or health promotion. Introduction: Introduction could be start with paragraph to review the framework and importance of health promotion and their objectives. The framework should be explain dimensions, strategies to guide their implementation. After that is important to describe factors related with implementation of health promotion, not only community visits are health promotion. After that, is necessary describe de program or health promotion strategy that implement in Spain, when it start, personnel involve, main objective, facilities involves, activities, previous evaluation and
--

	results about health promotion, the future goals. It is very important to describe the level on what health minister focalize the activities. Methods To select case and controls define as a criteria only if they performed community activities, but don't specify activities. Is necessary to describe activities the researchers define to select control? The primary health facilities has some characteristics like number of health personnel, productivity, epidemiological characteristics, and social determinants. Why researchers do not use same criteria to choose controls? What characteristics has group of experts? About sample size to the first study estimate teams, and the second study estimate professional, why is the reason? Why results of team are comparable with professional if the first refer to a group or persons and the second refer to an individual person. Researchers should describe dependent variable, how it was classify? Results  • Social worker are not comparable in cases and controls. Why you include in the analysis ? • Table 4 should be organize in a better way to understand the results.. Discussions It is a little confusing. Researchers must start the discussion with team characteristic and the comparability or not of cases and controls. In page ...line25 "This makes sense in the context of the Spanish health care system, which attempts to offer equitable care to people according to their needs and community activities are important tools to reduce inequalities in health care " I don't believe this text is centered in health promotion? Care is not the same that health promotion? Care has a curative perspective. Nurses are involved in community activities but need specify what kind of activities and link those with health promotion Conclusion Sample of social workers are insufficient in order to conclude in this study. Relevance of nurses in health promotion is indisputable References:  • Include references related to the framework of health promotion
--	---

REVIEWER	Jenni Judd Professor of Health Promotion School of Health, Medical and Applied Sciences Bundaberg Campus Central Queensland University Australia
REVIEW RETURNED	08-Feb-2017

GENERAL COMMENTS	This is an interesting study. The manuscript does need some editing for plain english to increase errors of explanation. There are a few definitions that could be inserted in the manuscript which would improve readability. I have made some suggested edits. I have not rechecked the statistics as I am not a statistician so someone should check these. I am not able to comment on all references as some of these are in spanish and I am not bilingual. There are some errors in references- eg Ottawa Charter is not referenced. Although I am not a statistician I think the results are interesting and add to this area of interest.
--

REVIEWER	Depeng jiang University of Manitoba, Winnipeg, Canada
REVIEW RETURNED	10-Mar-2017

GENERAL COMMENTS	This study examines factors associated with participation in health-promoting community activities. They examined factors at two different subject levels: one at teams, one at professionals. The article is potentially interesting to the readers of BMJ Open. My review mostly focus on the methodology/statistical analysis/results parts. There are several weaknesses as described below.  1. Line 44, Page 6. The second study examined teams that were cases in the first study. But these teams still differ each other. Without adjusting these team environment factors, it might be hard to 'adjust for community/team factors' to professionals who developed these activities (cases) versus those did not (controls). This is a big problem for second 'case-control' study. 2. Line 22-33, Page 9. It is not clear how OR of 2.5 and 1.75 were selected. Also you have power to detect each single predictor does not mean that you have power to conduct multivariable analyses (adjusted effect). So the sample size calculation here does not make sense to me. 3. I do not think conventional logistic regression is appropriate for second 'case-control' study. I does not agree with the author 'a multilevel analysis was not necessary'. I think you need to conduct multilevel analysis to examine factors at both team and professional level because of the hierarchical data structure (professionals nested in the team/community). Actually it might be more interesting
---

	to examine cross level impacts of these factors at different level, Overall, although the authors have some interesting empirical findings. Their work is simply not ready for publication because of drawback of study design and inappropriate statistical methods especially for second 'case-control' study.
--	--

REVIEWER	Catherine Best University of Stirling, UK
REVIEW RETURNED	13-Mar-2017

GENERAL COMMENTS	Overall the explanation of the data analysis is very good. The authors are very clear about the process they have used for the analysis by logistic regression and for each analysis this appears sound. My main concern would be the comparison across the two final adjusted models. The authors have conducted two separate analyses:  1. The first at the level of team where they have examined the team level factors that predict the participation of team in community health promoting activities and 2. the second analysis at the level of the health professional examining the factors predicting the participation of professionals in community health promoting activities (within teams that are involved in such activities). These two different models have different outcomes (one is team activity other is individual activity) different sample sizes and different numbers of predictors included in the model. I am not sure that directly comparing the model fit to conclude that individual factors are more important than team factors is appropriate. E.g in the section at top of pg 16 'The final model of community and team variables had poor explanatory power (Nagelkerke R squared = 0.11). In contrast, the model of the characteristics of professionals had a considerably better explanatory power (Nagelkerke R squared = 0.43). Thus, at least for the variables selected in this study, individual characteristics are more important for the development of health promoting community activities'. There is not a single outcome of health promoting activities – the two models examine different outcomes. I recommend avoiding direct comparison of the two models. In addition the interpretation of the Nagelkerke R squared as 'explanatory power' can be problematic. For logistic regression we only have pseudo R squared measures that do not have the same interpretation as the R squared of ordinary least squares regression. In OLS regression the R squared can be interpreted as the proportion of variance accounted for by the model. For logistic models there is no exact analogue. The Nagelkerke R squared is based on the ratio of the likelihoods of the intercept only model
--

	relative to the full model so is a measure of the improvement of the full model over the intercept only. With two different models with different 'intercept only' probabilities I do not think it makes sense to directly compare them based on the Nagelkerke R squared. Again my conclusion is to avoid the direct comparison. I agree that the model fit of the team model appears inferior to the individual level model but this is difficult to quantify and with different outcomes it is difficult to draw conclusions about what level factors have most impact on the performance of health promoting activities. A minor comment is that in the description of the analysis on page 9 it states that a 'saturated' model was constructed. In my understanding this refers to a model with as many parameters as data points. This however does not seem to be what the authors mean. Consider revising. The odds ratio for social workers in the professional level model is very wide due to having 1 social worker in one of the groups. No action – just noted. The headings within the tables need adjustment . Mean, p etc are in wrong columns in Table 1.
--	--

VERSION 1 – AUTHOR RESPONSE

Reviewer: 1

Reviewer Name: Alcalde-Rabanal J

Institution and Country: Institute of Public Health, Mexico

Please state any competing interests: No conflict of interest

Please leave your comments for the authors below

Title

Factors related to the development of health-promoting community activities in Spanish primary health care: Two case-control studies

3) I am not sure if it is the correct title, because in the paper I can't find a definition that the author use to classify community activities. If health personnel visit communities to follow a person who live with diabetes, it is not health promotion. This activity is preventive. In my opinion is important to describe with better detail community activities in order to verify if this paper is related with preventive framework or health promotion.

- We agree with the commentaries of the reviewer. The study is about Health promoting community activities. Community activities which have not a health promotion approach are not included. Neither health promotion interventions which have not a community approach. We change in the text every reference to community activities for "health promoting community activities (HPCA)" to avoid misunderstandings.

- The operative definition of Health promotion community activities (HPCA) was modified in the methodology section. The conceptual (and complete) definition was published in previous articles, like

the article of protocol which was attached to this one:

- Intervention and participation activities of which take place in groups showing common characteristics, needs or interests with the aim of promoting health, improving quality of life and social wellbeing, boosting the capacity of people and groups to solve their own problems, and meeting demands or needs in accordance with the following inclusion criteria:

- ♣ The health center professionals participate in CAs, regardless of whether they are the promoters or not.

- ♣ The activities have to be part of a program. Thus, they are not independent and isolated actions.

- ♣ The community actively participates in their design, implementation and/or assessment.

- ♣

- The following are excluded:

- ♣ Actions addressed only to diagnose specific health problems (screening programs, checkups, etc.) when the community is just a passive receiver.

- ♣ Actions addressed to the prevention of a specific health problem through the application of a specific therapy (such as vaccinations) when there is no active participation of a sector of the community.

- ♣ Actions addressed only to the detection of health risk factors and addictions.

- ♣ Specific individual visits to the healthcare provider, even though they are related to health promotion (family planning, giving up smoking, etc.), except in the case of being part of a broader project with the participation of a sector of the community.

- ♣ Actions addressed to groups, without contemplating the participation of a sector of the community in their organization, which are limited to the transfer of information such as talks or advice about healthy habits.

- We included the Ottawa definition of health promotion in the introduction. We included in methodology a specific comment about the exclusion of preventive interventions:

- Health preventive interventions which does not have a health promotion approach were excluded.

Introduction:

4) Introduction could be start with paragraph to review the framework and importance of health promotion and their objectives. The framework should be explain dimensions, strategies to guide their implementation. After that is important to describe factors related with implementation of health promotion, not only community visits are health promotion.

- OK. We do not pretend to address all the framework of health promotion in this article, although we agree it is interesting and important to add some more information, in addition to the references. We add the Ottawa definition, and some information in the first paragraph:

- The Ottawa Charter for Health Promotion¹ defined Health promotion as the process of enabling people to increase control over, and to improve, their health. This definition implies the importance of empowering people² to take control over what determines their health. One tool to work in this direction are health promoting community activities. These can come in a variety of forms, from walking groups in a neighbourhood to holding meetings with local authorities, schools or civil associations to develop specific health programs.

5) After that, is necessary describe de program or health promotion strategy that implement in Spain, when it start, personnel involve, main objective, facilities involves, activities, previous evaluation and

results about health promotion, the future goals. It is very important to describe the level on what health minister focalize the activities.

- We agree about the interest about the topic referred by the reviewer, but we have the space limitation and requirement of synthesis proper of the scientific publications. This paper is centered in primary health care attention level. This is very different between countries, so we try to summarize the relation of this level with health promotion in the second paragraph of the introduction:

Since the implementation of health care reform in Spain during the 1980's⁵⁻⁶, primary health care teams have had responsibility for carrying out health promotion activities. These teams, located in health care centres throughout the country, are composed of physicians, community nurses, paediatricians, midwives, social workers, and other health care professionals (gynaecologists, psychologists, and psychiatrists) in a system that offers almost-universal and free health care. These conditions are ideal for an interdisciplinary approach to encourage health promoting activities. However, as in other countries⁷, implementation of these interventions differs greatly among the different health care teams⁸⁻¹⁰, and relatively few professionals are involved¹⁰⁻¹².

- The main point that we try to explain to the reader is that primary health care in Spain have the legal responsibility and the resources to realize a health promoting community approach in his interventions, but they do not. Why? There are some structural and politics factors (not pointed in this paper), but also team, community and professional factors. In this manuscript we tried to focus in the team, community and individual professional factors. We think that if we start to explain the level of implication of the health minister in health promotion, our main point will be diluted. Also, health politics in Spain are not so easy to explain because they are not dependent of one unique ministry. Since more than twenty years ago, seventeen regions have autonomy to determine most of his health politics. That is why we think that explaining these aspects would make reading difficult and would contribute little to the purpose of the article. For more information about the topics discussed, we have another published article in Spanish which could be of interest, that is also referred in the current paper.

March S, Jordán M, Montaner I et al. ¿Qué hacemos en el barrio? Descripción de las actividades comunitarias de promoción de la salud en atención primaria: Proyecto frAC. Gac Sanit 2014; 28:267-73.

Methods

6) To select case and controls define as a criteria only if them performed community activities, but don't specify activities. Is necessary to describe activities the researchers define to select control? The primary health facilities has some characteristics like number of health personnel, productivity, epidemiological characteristics, and social determinants. Why researchers do not use same criteria to choose controls?

- We think may have been some misunderstanding of some aspect of the study. We performed two studies, both with case-control designs. In the first one we use health care teams like reference units. Cases were teams who had performed HPCA in the previous year. Controls were teams who had not performed HPCA. So, controls did not performed any HPCA.

- The second study, was performed only within the health teams who had participated in a HPCA in the previous years (the cases of the first study). Cases were the professionals involved in those HPCA. Controls were professionals in the same health team who decided not to be involved in this HPCA. So, they did not perform HPCA. This controls were randomly selected from the team

professionals who did not carry HPCA and who were in the same team of the cases recruited.

- The characteristics of the teams, primary health care facilities and community had no relation to the selection of cases and controls, but are already results obtained from the study.

- We made some changes in the "selection of cases and controls" section to avoid misunderstandings:

We used a conceptual definition of health promoting community activity reached in a previous study,³⁰⁻³¹ which was converted into an algorithm for use in this study²⁹. In particular, this activity had to be a non-isolated activity carried out in the previous year, in which the professional participated on behalf of the health centre. Furthermore, it had to have involvement and active participation of the community, or the population had to have the capacity to influence the intervention, or it was a cross-sector activity involving collaboration with entities outside of health care (e.g. education, social services). Health preventive interventions which does not have a health promotion approach were excluded.

We telephoned all the health centres in the research areas asking if they had participated in community activities during the last year. If they did, we contacted a participant of the activity and gave him/her a questionnaire to confirm that the activity met our inclusion criteria. For the first study, health teams that participated in at least one community activity in the last year were selected as cases, and controls were those who had not participated. For the second study, we selected all professionals who participated in the implementation of those community activities as cases and professionals who did not participate in the planning as controls. In every health team we selected at least one control for every case.

7) What characteristics has group of experts?

- This group of experts was defined in a previous work, published in Spanish and referred in the text.

March S, Ramos M, Soler M, Ruiz-Jiménez JL, Miller F, Domínguez J, et al: Revisión documental de experiencias de actividad comunitaria en atención primaria de salud. *Aten Primaria* 2011, 43:289-296.

- The group was part of a scientific society and was a multidisciplinary team involving physicians, nurses, social workers and sociologists working in primary health and public health care. We think this information it is not so relevant to include it in this long article. The reader with interest in such details could follow the references. But we change the text to clarify:

We used a conceptual definition of health promoting community activity reached in a previous study 30-31.

8) About sample size to the first study estimate teams, and the second study estimate professional, why is the reason? Why results of team are comparable with professional if the first refer to a group or persons and the second refer to an individual person.

- We performed two studies with two different unities of reference: In the first one the health care teams, in the second one the professionals. This is the reason why we needed two different estimates of sample size.

- We agree with the reviewer that the results of the models are not comparable. We change the text to avoid the comparisons. We maintain those comparisons that are based on interpretations of the research team and not comparisons of statistical parameters. In this regard, like the reviewer, the research team maintains that the interpretation of the results implies that individual factors have an

important value with respect to those of the team/community in carrying out health promoting community activities.

9) Researchers should describe dependent variable, how it was classify?

- Like in all the case controls designs, the dependent variable is to be case or to be control. So, for the first study the variable is teams involved (or not) in HPCA. For the second the variable is professionals involved (or not) in HPCA.

- This variables were well identified in the results:

- The variables significantly related to team involvement in community activities were...

- Table 2 shows that the variables related to involvement in community activities were...

- The final multivariate model (Table 4) shows that the following factors were significantly associated with health-promoting community activities:

- Following the reviewer's warning and to avoid confusion, we include this phrase in the text of the analysis plan:

Main variable was, for both studies, to be case or control.

Results

10) Social worker are not comparable in cases and controls. Why you include in the analisis ?

- Case-Controls design is not like clinical trials, where all is about comparability between groups. Precisely because we are looking for factors (expositions), we focus in the differences between groups. This is one study exploratory (no previous studies about this specific topic), and one of the advantages of the case-controls designs is the possibility to asses different factors at the same time. Controls were randomly selected within the professionals in the health care teams who were not cases. Then, if they are differences between cases and controls in the factors it is because this factors are relevant for the purpose of the study. This is so in the case of the profession, particularly for social workers. So, our interpretation of the results indicates that there were so few social workers controls randomly selected because the majority were involved in HPCA. If being a social worker was not an associated factor, there would be the same in both groups. The statistical data indicates there are enough to do comparisons (Coefficient interval of the adjusted OR are greater than one).

-In Spain every primary health care team has a social worker. In some regions they are the link of the team with the community, and they have a very important health promoter role. In others regions is not like this. So it's very important for our study to maintain the social workers in the sample, both for theoretical and for statistical reasons.

11) Table 4 should be organize in a better way to understand the results..

- OK. Following the advice, we changed the Table 4.

Discussions

12) It is a little confusing. Researchers must start the discussion with team characteristic sand the comparability or not of cases and controls.

- Case-Controls design is not like clinical trials, where all is about comparability between groups.

Precisely because we are looking for factors (expositions), we focus in the differences between groups. Problem with case-controls could be in the selection of controls, not in comparability between groups. In this studies that was not a problem, because controls are randomly selected in between the target population.

- Any case, we have modified the first part of the discussion, focusing on the main results.

13) In page ...line25 "This makes sense in the context of the Spanish health care system, which attempts to offer equitable care to people according to their needs and community activities are important tools to reduce inequalities in health care " I don't believe this text is centered in health promotion? Care is not the same that health promotion? Care has a curative perspective.

- We agree with the reviewer but we think there is a misunderstanding because of the translation. We are talking here about general attention of the health care system. Equal attention is one of the fundamental principles of the system. Regardless of whether this attention is curative, palliative, preventive or health promotion. We changed word "care" for "attention" to avoid misunderstandings.

14) Nurses are involved in community activities but need specify what kind of activities and link those with health promotion

- This would be very interesting and in fact we published some results about this topic. But this is not the objective of the present article. If it is of the reviewer (or the reader, because has been referred in the text) interest, information can be found here:

March S, Jordán M, Montaner I et al. ¿Qué hacemos en el barrio? Descripción de las actividades comunitarias de promoción de la salud en atención primaria: Proyecto frAC. Gac Sanit 2014; 28:267-73.

- We included this in the text:

Description of health promoting community activities are described elsewhere³⁰.

Conclusion

15) Sample of social workers are insufficient in order to conclude in this study. Relevance of nurses in health promotion is indisputable

- We do not agree in this with the reviewer. We have enough sample of social workers to develop a multivariate analysis. Besides, the adjusted model notes a very high OR for social workers regarding nurses. And the confidence intervals were significant. Also we had theoretical reasons to maintain the social workers because their relevance in our context.

May be the reviewer know something about health promotion in primary health care in Spain that leads her to think that nurses play a much more important role than social workers, but our data do not indicate that, and so we cannot change it.

References:

16) Include references related to the framework of health promotion

- OK. We included more references about the general framework and concepts of health promotion..

Reviewer: 2

Reviewer Name: Jenni Judd

Institution and Country: Professor of Health Promotion, School of Health, Medical and Applied

Sciences, Bundaberg Campus, Central Queensland University, Australia
Please state any competing interests: None declared

Please leave your comments for the authors below

17) This is an interesting study. The manuscript does need some editing for plain english to increase errors of explanation. There are a few definitions that could be inserted in the manuscript which would improve readability. I have made some suggested edits. I have not rechecked the statistics as I am not a statistician so someone should check these. I am not able to comment on all references as some of these are in spanish and I am not bilingual. There are some errors in references- eg Ottawa Charter is not referenced. Although I am not a statistician I think the results are interesting and add to this area of interest.

- We greatly appreciate the comments of the reviewer. We had reviewed and made changes to the text following her advice. We also include the references mentioned.

Reviewer: 3

Reviewer Name: Depeng jiang

Institution and Country: University of Manitoba, Winnipeg, Canada

Please state any competing interests: None declared

Please leave your comments for the authors below

18) The article is potentially interesting to the readers of BMJ Open.

- We greatly appreciate the comments of the reviewer

My review mostly focus on the methodology/statistical analysis/results parts. There are several weaknesses as described below.

19) Line 44, Page 6. The second study examined teams that were cases in the first study. But these teams still differ each other. Without adjusting these team environment factors, it might be hard to 'adjust for community/team factors' to professionals who developed these activities (cases) versus those did not (controls). This is a big problem for second 'case-control' study.

- The second study was developed only in the teams whose had done health promotion community activities. So in the same center we recruited some cases and some controls. Then, the team/community factors are adjusted because are the same in cases and in controls. That is the point of this design. Case controls are matched (adjusted by design) in the same team. We made some changes in the text to clarify this point.

The second study examined only teams that implemented community activities to adjust for community/team factors;

- This is indicated in the Strength and limitations section:

Furthermore, our design of the second study, in which cases and controls were matched by teams, adjusted for the effect of team and community variables, so a multilevel analysis was not necessary.

20). Line 22-33, Page 9. It is not clear how OR of 2.5 and 1.75 were selected. Also you have power to detect each single predictor does not mean that you have power to conduct multivariable analyses (adjusted effect). So the sample size calculation here does not make sense to me.

- These OR point the statistical power achieved with these sample sizes. These are one of the variable parameters in the formula for calculating sample size in case-control studies.

SK Lwanga, S Lemeshow. Sample size determination in health studies: a practical manual. Geneva: World Health Organization; 1991.

- This calculation of sample size for case -controls studies works to develop multivariable analyses, so no adjusted effect is necessary. We did not find studies that applied a design effect in the calculation of the sample size by performing multivariate analysis. We could find some that applies it because they performed multilevel analysis, which we have not done precisely because the data in the second level (team/community) were adjusted by design (they are the same in cases and in controls).

- This OR threshold is determined by the research team, based in knowledge and literature. Normally this means clinical relevance. But in this health services research without previous quantitative studies, the OR were determined by the research team. To classify this, we added the following to the sample size calculation section:

Since there is no previous research on this topic, the ORs were decided by the research team based on relevancy and feasibility criteria.

- In case control studies reported in this same journal this is the standard way of presenting the sample size calculation (when there is any). Some examples:

<http://bmjopen.bmj.com/content/5/2/e006355>

<http://bmjopen.bmj.com/content/4/4/e004499>

<http://bmjopen.bmj.com/content/3/10/e003871>

21). I do not think conventional logistic regression is appropriate for second 'case-control' study. I does not agree with the author 'a multilevel analysis was not necessary'. I think you need to conduct multilevel analysis to examine factors at both team and professional level because of the hierarchical data structure (professionals nested in the team/community). Actually it might be more interesting to examine cross level impacts of these factors at different level.

- We agree with the reviewer that it might be interesting to examine cross level impacts of these factors, but no with this data and this design. Multilevel analysis was impossible to conduct with our data. As already explained, the second study was developed ONLY within the teams who were cases on the first study. In every team we selected cases and almost same number of controls. The point here is that Cases and controls have the same values of team/community factors because they are from the same teams and communities. So, there is not possible comparability, and there is not possibility of carry a multilevel analysis. This matching strategy is commonly used in case control studies, and is a way to adjust by design.

22) Overall, although the authors have some interesting empirical findings. Their work is simply not ready for publication because of drawback of study design and inappropriate statistical methods especially for second 'case-control' study.

-We think that with the answers to the reviewer, and the modifications made to avoid misunderstandings, It is possible to evaluate the article again.

Reviewer: 4

Reviewer Name: Catherine Best

Institution and Country: University of Stirling, UK

Please state any competing interests: None declared

Please leave your comments for the authors below

23) Overall the explanation of the data analysis is very good. The authors are very clear about the process they have used for the analysis by logistic regression and for each analysis this appears

sound. My main concern would be the comparison across the two final adjusted models. The authors have conducted two separate analyses:

1. The first at the level of team where they have examined the team level factors that predict the participation of team in community health promoting activities and
2. the second analysis at the level of the health professional examining the factors predicting the participation of professionals in community health promoting activities (within teams that are involved in such activities).

These two different models have different outcomes (one is team activity other is individual activity) different sample sizes and different numbers of predictors included in the model. I am not sure that directly comparing the model fit to conclude that individual factors are more important than team factors is appropriate. E.g. in the section at top of pg 16 'The final model of community and team variables had poor explanatory power (Nagelkerke R squared = 0.11). In contrast, the model of the characteristics of professionals had a considerably better explanatory power (Nagelkerke R squared = 0.43). Thus, at least for the variables selected in this study, individual characteristics are more important for the development of health promoting community activities'. There is not a single outcome of health promoting activities – the two models examine different outcomes. I recommend avoiding direct comparison of the two models.

In addition the interpretation of the Nagelkerke R squared as 'explanatory power' can be problematic. For logistic regression we only have pseudo R squared measures that do not have the same interpretation as the R squared of ordinary least squares regression. In OLS regression the R squared can be interpreted as the proportion of variance accounted for by the model. For logistic models there is no exact analogue. The Nagelkerke R squared is based on the ratio of the likelihoods of the intercept only model relative to the full model so is a measure of the improvement of the full model over the intercept only. With two different models with different 'intercept only' probabilities I do not think it makes sense to directly compare them based on the Nagelkerke R squared. Again my conclusion is to avoid the direct comparison. I agree that the model fit of the team model appears inferior to the individual level model but this is difficult to quantify and with different outcomes it is difficult to draw conclusions about what level factors have most impact on the performance of health promoting activities.

- We appreciate and fully agree with the reviewer's comments. So, we remove from the text the comparative assessments that the reviewer points out. We maintain those comparisons that are based on interpretations of the research team and not comparisons of statistical parameters. In this regard, like the reviewer, the team maintains that the interpretation of the results implies that individual factors have an important value with respect to those of the team/community in carrying out health promoting community activities.

24) A minor comment is that in the description of the analysis on page 9 it states that a 'saturated' model was constructed. In my understanding this refers to a model with as many parameters as data points. This however does not seem to be what the authors mean. Consider revising.

- Ok. Agree. We change the text:

- We constructed a first model using each variable whose p-value was less than 0.2 in the bivariate analysis

25) The odds ratio for social workers in the professional level model is very wide due to having 1 social worker in one of the groups. No action – just noted.

- OK. Agree. But control groups were randomly selected, and the cases intentionally. So in controls this 1 is a representation of his proportion in population who do not community activities. And in cases are overrepresented because they do more health promoting community activities. This explains the

odd ratio.

26) The headings within the tables need adjustment . Mean, p etc are in wrong columns in Table 1.

VERSION 2 – REVIEW

REVIEWER	Jenni Judd School of Health Medical and Social Sciences Central Queensland University Bundaberg QLD Australia
REVIEW RETURNED	09-May-2017

GENERAL COMMENTS	Thanks for the opportunity to review this paper. This paper has improved and addressed the concerns I raised in the first review. There are a couple of edits that I have suggested, just minor. An interesting read.
---

REVIEWER	Depeng Jiang University of Manitoba, Canada
REVIEW RETURNED	08-May-2017

GENERAL COMMENTS	1. My previous comments: The second study examined teams that were cases in the first study. But these teams still differ each other. Without adjusting these team environment factors, it might be hard to 'adjust for community/team factors' to professionals who developed these activities (cases) versus those did not (controls). This is a big problem for second 'case-control' study. I do not agree with authors' justification that multilevel analysis was not necessary. Traditional multiple regression techniques including logistic regression treat the units of analysis as independent observations. One consequence of failing to recognize hierarchical structures is that standard errors of regression coefficients will be underestimated, leading to an overstatement of statistical significance. Standard errors for the coefficients of higher-level predictor variables will be the most affected by ignoring grouping. In a multilevel model the groups in the sample (here health promotion communities) are treated as a random sample from a population of groups. Using a fixed effects model, inferences cannot be made beyond the health promotion communities in the sample. Unless authors can justify that the intraclass correlation is zero or very low, I think multilevel analyses are better approach. 2. Line 22-33, Page 9. It is not clear how OR of 2.5 and 1.75 were selected. Also you have power to detect each single predictor
--

	does not mean that you have power to conduct multivariable analyses (adjusted effect). So the sample size calculation here does not make sense to me. Thanks for the authors resolved my this concern. 3. I do not think conventional logistic regression is appropriate for second 'case-control' study. I does not agree with the author 'a multilevel analysis was not necessary'. I think you need to conduct multilevel analysis to examine factors at both team and professional level because of the hierarchical data structure (professionals nested in the team/community). Actually it might be more interesting to examine cross level impacts of these factors at different level, See my Comment 1. You do not need to have control teams/communities (controls in study 1) to conduct multilevel analysis. Even you match cases with controls (professionals) within the same team/community. If it is matched beside the community/team factors, you should conduct conditional logistic regression for case-control matched study. So overall my comment is still "Their work is simply not ready for publication because of drawback of study design and inappropriate statistical methods especially for second 'case-control' study."
--	--

REVIEWER	Dr Catherine Best University of Stirling, Faculty of Health Sciences and Sport
REVIEW RETURNED	10-May-2017

GENERAL COMMENTS	Response to revision of bmjopen-2017-015934.R1 Factors related to the development of health-promoting community activities in Spanish primary health care: Two case-control studies The authors have greatly improved the manuscript. In particular, the attention to readability and English language editing have been very effective. I think the manuscript requires a little more revision before publication. Firstly, although I agree that the model of individual influences on health promotion activity is a better model than the team model of health promotion activity – I do not think these can be directly compared to deduce which level (individual or team) is most important for predicting health promotion activity. The first model (the team model) tells us which team factors are most important for predicting whether or not a primary care team will undertake health promotion activity. The second model tells us, within a team that undertakes health promotion activity, who is the most likely to be the person(s) to do the health promotion activity. These are not comparable outcomes. The relative utility of the two models could be compared in the discussion but I think making strong conclusions (for example in the abstract) that 'Professional characteristics seem to have a greater influence than team/community factors in performing health-promoting community activities' is not supported by these analyses. A multi-level model would allow comparison of the effects at different levels. However I imagine that there are very few professionals per team in the second analysis (exact number per team is not stated).
---

	There are 597 professionals in the second part and 103 teams that do health promotion activities so number per team must be less than six which is too few units per cluster for random effects modelling. In table 2 the heading 'p*' is on the wrong column.
--	--

VERSION 2 – AUTHOR RESPONSE

Reviewer: 3

Reviewer Name: Depeng Jiang

Institution and Country: University of Manitoba, Canada

Please state any competing interests: None declared

This revision of the manuscript has not resolved my major concerns with statistical analyses. I appreciate the authors' attention and response to my prior comments. The article is potentially interesting to the readers of BMJ Open. My review mostly focus on the methodology/statistical analysis/results parts. There are several weaknesses as described below.

1. My previous comments: The second study examined teams that were cases in the first study. But these teams still differ each other. Without adjusting these team environment factors, it might be hard to 'adjust for community/team factors' to professionals who developed these activities (cases) versus those did not (controls). This is a big problem for second 'case-control' study. I do not agree with authors' justification that multilevel analysis was not necessary.

Traditional multiple regression techniques including logistic regression treat the units of analysis as independent observations. One consequence of failing to recognize hierarchical structures is that standard errors of regression coefficients will be underestimated, leading to an overstatement of statistical significance. Standard errors for the coefficients of higher-level predictor variables will be the most affected by ignoring grouping. In a multilevel model the groups in the sample (here health promotion communities) are treated as a random sample from a population of groups. Using a fixed effects model, inferences cannot be made beyond the health promotion communities in the sample. Unless authors can justify that the intraclass correlation is zero or very low, I think multilevel analyses are better approach.

RESPONSE TO THE REVIEWER:

As we commented in the previous draft, we would love to examine cross level impacts of these factors but we cannot for some of these reasons:

- We agree with the statements about the convenience of using the multilevel models cited by the reviewer from the Bristol university (We also did their fantastic online courses when we started modeling multilevel) But the fact is that with our data these reasons are not fulfilled. As the Bristol University says (<http://www.bristol.ac.uk/cmm/learning/multilevel-models/what-why.html>):

- . One consequence of failing to recognize hierarchical structures is that standard errors of regression coefficients will be underestimated, leading to an overstatement of statistical significance. Standard errors for the coefficients of higher-level predictor variables will be the most affected by ignoring grouping.

But we can demonstrate that factors at the team/community level will have almost no effect with this team/community matched design. That is commonly the purpose of the matching.

- Case and controls were selected from the same teams. This means they have approximately the same values on average of the variables at the team/community level. If for example, the cases in the team 1 worked in a medium-low socioeconomic level communities, the controls in the same team

would work in the same kind of communities (in fact, the same). This makes that the total proportion of cases that work in this kind of communities in cases, would be almost the same as in controls. This is because we selected more or less one control by case in every team. Differences would be explained only by this differences in selection (for example, if we selected one case and two controls in one team), not because they are really factors to be considered. So if we considered these factors in the analysis with a multilevel approach, we would not expect a significative team effect which fits in the model.

- This does not mean that there are not hierarchical effects. Same paper indicates that community/team factor have some impact in developing health promotion community activities. At least in primary care teams involvement in these activities. But it means that our results are adjusted by these effects. They are controlled because with the matched selection, they have approximately the same values for case and for controls.

- If the controls would had been randomly selected from any other center, the multi-level analysis could have been performed. Although this approach would have complicated the data collection (we are talking about a study that covers 5 Spanish regions). We have chosen the classical selection procedure adapting to our possibilities and context: the matching with factors of the team/community.

- The advantages, in terms of feasibility and efficiency, of the chosen design are obvious, and in any case, in our opinion, it is not a so bad design in the sense that there is low risk of bias in the data we obtained for the study question. The question that tries to answer the design of the second study is: regardless of the community/team factor (that means, if they were almost the same), what factors (individual) differentiate professionals who perform health promoting community interventions from those who do not?

- We believe that our results are robust to answer that question, and also they have been confirmed by other statisticians with whom we have consulted.

- Moreover, we selected 597 professionals from 103 teams. That means less than 6 professionals on average: approximately half cases and half controls. As indicated by another reviewer, that would be too few units per cluster for random effects modeling.

- However, obviating all of these limitations of design and sample cluster size, we tried to construct a binary generalized linear model with mixed effects, attributing random effects at the team/community level, and the results were similar to those of the model presented in the article: Team/community factors did not fit the model, as expected, and also individual factors had no fundamental changes.

- We believe that our results and analysis are solid as they are. No hierarchical modeling approach is possible with this matched case controls selection. The results of the second study have to be interpreted as "adjusted by team/community factors".

REVIEWER'S COMMENT

2. Line 22-33, Page 9. It is not clear how OR of 2.5 and 1.75 were selected. Also you have power to detect each single predictor does not mean that you have power to conduct multivariable analyses (adjusted effect). So the sample size calculation here does not make sense to me.

Thanks for the authors resolved my this concern.

3. I do not think conventional logistic regression is appropriate for second 'case-control' study. I does not agree with the author 'a multilevel analysis was not necessary'. I think you need to conduct multilevel analysis to examine factors at both team and professional level because of the hierarchical

data structure (professionals nested in the team/community). Actually it might be more interesting to examine cross level impacts of these factors at different level, See my Comment 1. You do not need to have control teams/communities (controls in study 1) to conduct multilevel analysis. Even you match cases with controls (professionals) within the same team/community. If it is matched beside the community/team factors, you should conduct conditional logistic regression for case control matched study.

RESPONSE TO THE REVIEWER:

We agree that the conditional logistic regression approach is a procedure to be evaluated in matched case-control studies, although it is an option that has some controversy and is not always the most recommended (Greenland 2000, Pearce 2016).

- In the sources consulted (Hansson 2001), it is suggested that when the pairing is proportional (close to 1: 1, as in our case), non-conditional logistic regression offers better adjustments with lower bias. Not so in more unequal pairings, like 1: 4 or more.

- Conditional logistic regression may be useful in paired studies of reduced sample size, or with a large number of parameters relative to the total number of subjects (Rahman 2003). It is demonstrated that in the case control designs, when the probabilities of choosing the individuals in both groups are independent of the independent variables of the model, as in our case, the coefficient estimates produce the same results through the likelihood functions Conditional or non-conditional. As this condition is difficult to prove, so in practice unconditional logistic regression is used when the sample size is large or the number of parameters is small relative to the total number of subjects, as it is in this study.

-

Pearce N. Analysis of matched case-control studies. *BMJ* 2016;352:i969

Hansson L., 2001. Statistical Considerations in the Analysis of Matched Case-Control Studies. With Applications in Nutritional Epidemiology. Acta Universitatis Upsaliensis. Comprehensive Summaries of Uppsala Dissertations from the Faculty of Social Sciences 100. 33 pp. Uppsala. ISBN 91-554-4950-6.

M. Rahman, J. Sakamoto, T. Fukui. Conditional versus unconditional logistic regression in the medical literature. *J Clin Epidemiol* 2003;56:101-102

Greenland S, Schwartzbaum JA, Finkle WD. Problems due to Small Samples and Sparse Data in Conditional Logistic Regression Analysis. *Am J Epidemiol* 2000;151:531–9.

REVIEWER'S COMMENT

So overall my comment is still "Their work is simply not ready for publication because of drawback of study design and inappropriate statistical methods especially for second 'casecontrol' study."

RESPONDE TO THE REVIEWER:

We regret not to agree with this reviewer comment.

- Our article tries to answer two relevant and different questions that have not been answered previously in the literature.

- The first is: what factors are related to primary health care teams to carry on health promotion community interventions? This question and the design, analysis and results that answer it have not received criticism from the reviewer, who just mentioned that it was interesting, which we appreciate.

- In the second, we wondered: assuming that these factors of community/team were the same, that is, from the same health center, which causes primary care professionals to be involved in this health promoting community activities and others not? This question we think is scientifically relevant and is answered in the best possible way adapted to our context and our possibilities. The results are solid to answer that question. There is no known bias.
- There is no criticism by the reviewer about the relevance of these research questions.
- We think that the reviewer raises another research question, which we think can be very interesting and relevant, but that cannot be answered with our data as we have collected them. This would be: what is the weight of the factors of the team with respect to the individual factors in the accomplishment of community activities by the primary care professionals at the individual level?
- If someone ever answers that question, we are sure that the results of this study will be of great interest to them as they are presented in this article.
- We look forward to answer this question in future studies, but we cannot do it with this data.
- We kindly ask the reviewer to reconsider his position in light of the above.

Reviewer: 2

Reviewer Name: Jenni Judd

Institution and Country: School of Health Medical and Social Sciences, Central Queensland University, Bundaberg QLD Australia

Please state any competing interests: None declared

Please leave your comments for the authors below

Thanks for the opportunity to review this paper. This paper has improved and addressed the concerns I raised in the first review. There are a couple of edits that I have suggested, just minor. An interesting read.

RESPONDE TO THE REVIEWER:

We appreciate the contributions and comments of the reviewer, which undoubtedly improve the quality of the article.

Reviewer: 4

Reviewer Name: Dr Catherine Best

Institution and Country: University of Stirling, Faculty of Health Sciences and Sport

Please state any competing interests: No competing interests

Please leave your comments for the authors below

Response to revision of bmjopen-2017-015934.R1

Factors related to the development of health-promoting community activities in Spanish primary health care: Two case-control studies

The authors have greatly improved the manuscript. In particular, the attention to readability and English language editing have been very effective.

RESPONDE TO THE REVIEWER:

We appreciate the reviewer comments.

REVIEWER'S COMMENT

I think the manuscript requires a little more revision before publication. Firstly, although I agree that the model of individual influences on health promotion activity is a better model than the team model of health promotion activity – I do not think these can be directly compared to deduce which level

(individual or team) is most important for predicting health promotion activity. The first model (the team model) tells us which team factors are most important for predicting whether or not a primary care team will undertake health promotion activity. The second model tells us, within a team that undertakes health promotion activity, who is the most likely to be the person(s) to do the health promotion activity. These are not comparable outcomes. The relative utility of the two models could be compared in the discussion but I think making strong conclusions (for example in the abstract) that 'Professional characteristics seem to have a greater influence than team/community factors in performing health-promoting community activities' is not supported by these analyses.

RESPONDE TO THE REVIEWER:

We really appreciate the comprehension of the reviewer and we agree with the comments. We change conclusions in abstract and text to follow them.

ABSTRACT

Conclusions: Professional personal characteristics, as social sensitivity, profession, to feel team support, or motivation, have influence in performing health-promoting community activities. In contrast to the opinion expressed by many professionals, workload is not related to performance of health-promoting community activities.

TEXT

Conclusions

A key responsibility of primary health care is provision of health-promoting community activities. We found that the actual implementation of these activities depends more on the characteristics of individual professionals and on some characteristics of health care team or community than on his professional responsibility like primary health care workers. There were also significant differences among different types of professionals, and the contributions of nurses and social workers are fundamental. The pressures perceived by members of the health care team were not associated with involvement in community activities.

A multi-level model would allow comparison of the effects at different levels. However I imagine that there are very few professionals per team in the second analysis (exact number per team is not stated). There are 597 professionals in the second part and 103 teams that do health promotion activities so number per team must be less than six which is too few units per cluster for random effects modelling.

RESPONDE TO THE REVIEWER:

We agree and we appreciate the reviewer's comment

REVIEWER'S COMMENT

In table 2 the heading 'p*' is on the wrong column.

VERSION 3 - REVIEW

REVIEWER	Jenni Judd Health Medical and Applied Sciences Central Queensland University, Bundaberg Qld
REVIEW RETURNED	27-Jun-2017

GENERAL COMMENTS	The manuscript has improved however there are still a number of areas within the manuscript that need a little attention. I have attached it for your information. I am not able to comment on the
--

	statistical analysis so I will leave that to one of the other reviewers. There are some issues related to the limitations that could be improved and I have made suggestions in this regard. The reviewer also provided a marked copy with additional comments. Please contact the publisher for full details.
--	--

REVIEWER	Depeng Jiang University of Manitoba, Canada
REVIEW RETURNED	16-Jun-2017

GENERAL COMMENTS	All my previous concerns were addressed. If authors can add a short discussion about why multilevel analyses were not required, that will be great.
---

REVIEWER	Catherine Best University of Stirling, UK
REVIEW RETURNED	09-Jun-2017

GENERAL COMMENTS	I am happy for this to be published. From an English language point of view there are some small areas that could be rephrased. E.g just above Conclusions- 'undergraduate sanitary formation' is not a commonly used phrase. Maybe use 'training of healthcare workers'. In the conclusions section the sentence 'We found that the actual implementation...' is rather long and the end does not make sense. Consider deleting the end section 'than on his professional responsibility like healthcare workers'.
--

VERSION 3 – AUTHOR RESPONSE

We deeply appreciate the effort and patience of the reviewers and editorial team with our manuscript, which has certainly improved thanks to their work.

- We reviewed the text and we made some changes as reviewers suggested.
- We change the conclusions as the reviewer 4 suggested.
- As suggested by the reviewer 3, we included this sentence in discussion: Furthermore, our design of the second study adjusted the effect of team and community variables matching cases and controls by teams, so a multilevel analysis was not necessary.
- We also made the changes suggested by the reviewer 2 in the manuscript.

VERSION 4 - REVIEW

REVIEWER	Jenni Judd Central Queensland University Bundaberg, Queensland Australia
REVIEW RETURNED	04-Jul-2017

GENERAL COMMENTS	This manuscript is much improved and more easily replicated due to better descriptions of methods etc. My suggestions are mainly editorial: These are coloured in red. on page 3 in the abstract;p.4 in strengths and limitations; p.6 line 23; p.7 line 3 american spelling of center where all others are spelt centre; line 56 delete from; p.9 line 23 NGO needs spelling out; p 20 line 29 the use of sanitarian professional appears to be the incorrect term. This is the first time this is mentioned. p.20 line 50 insert such before as.
---

VERSION 4 – AUTHOR RESPONSE

We have made the changes suggested by the reviewer and the editorial team. We appreciate your patient work.